# Potential short-term negative versus positive effects of olive mill-derived biochar on nutrient availability in a calcareous loamy sand soil

Azzaz Alazzaz[1], Adel R. A. Usman[1,2], Munir Ahmad [1], Hesham M. Ibrahim[1,3], Jamal Elfaki[1,4], Abdelazeem S. Sallam[1], Mutair A. Akanji [1], Mohammad I. Al-Wabel [1,5]*

1 Soil Sciences Department, College of Food & Agricultural Sciences, King Saud University, Riyadh, Kingdom of Saudi Arabia, 2 Department of Soils and Water, Faculty of Agriculture, Assiut University, Assiut, Egypt, 3 Department of Soils and Water, Faculty of Agriculture, Suez Canal University, Ismailia, Egypt, 4 Nile Valley University, River Nile State, Sudan, 5 Department of Science and Environmental Studies, The Education University of Hong Kong, Hong Kong, Hong Kong

* malwabel@ksu.edu.sa

**Data Availability Statement:** All excel and data files are available from the DSR (scientific

## Abstract

In the present work, the olive mill solid waste (OMSW)-derived biochar (BC) was produced at various pyrolytic temperatures (300–700˚C) and characterized to investigate its potential negative versus positive application effects on pH, electrical conductivity (EC), and nutrients (P, K, Na, Ca, Mg, Fe, Mn, Zn, and Cu) availability in a calcareous loamy sand soil. Therefore, a greenhouse pot experiment with maize (*Zea mays* L.) was conducted using treatments consisting of a control (CK), inorganic fertilizer of NPK (INF), and 1% and 3% (w/w) of OMSW-derived BCs. The results showed that BC yield, volatile matter, functional groups, and zeta potential decreased with pyrolytic temperature, whereas BC pH, EC, and its contents of ash and fixed carbon increased with pyrolytic temperature. The changes in the BC properties with increasing pyrolytic temperatures reflected on soil pH, EC and the performance of soil nutrients availability. The BC application, especially with increasing pyrolytic temperature and/or application rate, significantly increased soil pH, EC, $NH_4OAc$-extractable K, Na, Ca, and Mg, and ammonium bicarbonate-diethylenetriaminepentaacetic acid (AB-DTPA)-extractable Fe and Zn, while AB-DTPA-extractable Mn decreased. The application of 1% and 3% BC, respectively, increased the $NH_4OAc$-extractable K by 2.5 and 5.2-fold for BC300, by 3.2 and 8.0-fold for BC500, and by 3.3 and 8.9-fold for BC700 compared with that of untreated soil. The results also showed significant increase in shoot content of K, Na, and Zn, while there was significant decrease in shoot content of P, Ca, Mg, and Mn. Furthermore, no significant effects were observed for maize growth as a result of BC addition. In conclusion, OMSW-derived BC can potentially have positive effects on the enhancement of soil K availability and its plant content but it reduced shoot nutrients, especially for P, Ca, Mg, and Mn; therefore, application of OMSW-derived BC to calcareous soil might be restricted.

publication portal) database (accession numbers
1493-043).

**Funding:** Author who received each award:
Mohammad Alwabel Fund Number:RG-1439-043
Funder: Deanship of Scientific Research, King Saud
University URL: http://dsrs.ksu.edu.sa The funders
had no role in study design, data collection and
analysis, decision to publish, or preparation of the
manuscript.

**Competing interests:** The authors have declared
that no competing interests exist.

## Introduction

Rapid expansion in agriculture to feed the continuously growing world population has
increased conventional intensification in farming systems, which consequently has resulted in
soil nutrient depletion and various environmental concerns [1]. Intensive application of agro-
chemicals has resulted in land degradation as well as declined soil health and quality [2, 3].
These problems are even more intensified in calcareous soils under arid and semi-arid climate
due to lower organic matter and nutrient availability. Hence, a global transition towards mod-
ern farming systems with sustainable soil health, safe ecosystems, food security, and climate
change mitigation is required. In this context, organic soil additives can potentially improve
crop production, soil organic matter, rehabilitation of degraded land, and microbial activity
with minimal environmental damage [4–7]. Biochar has been recently suggested as an emerg-
ing organic conditioner that can aid in overcoming soil problems and enhancing soil produc-
tivity [8–10].

Biochar is produced from organic materials (e.g., biomass) through pyrolytic processes
under limited oxygen supply, and is generally characterized by its high content of fixed carbon
[11]. Biochar materials are largely employed as additives to overcome soil problems and limita-
tions by enhancing soil properties in relation to chemical, biological, and hydro-physical
parameters, as well as nutrient content and efficiency [8, 10, 12–14]. Owing to higher cation
exchange capacity, sorption affinity, large surface area, higher porosity, and lower mobile mat-
ter, biochar could serve as an ideal candidate for improving physio-chemical properties of soils
such as cation exchange capacity, pH, water holding capacity, and nutrient retention [15].
However, previous studies have demonstrated that application of biochar was more suitable
for acidic soils than the alkaline soils, owing to liming effect induced by the alkaline nature of
biochar [16, 17]. Nevertheless, some studies demonstrated that the application of biochar to
calcareous sandy soil increases water holding capacity, organic matter, cation exchange capac-
ity, and microbial activities, whereas, decreases the hydraulic conductivity [18–20]. Amin [21]
reported that application of biochar to calcareous sandy soil improved phosphorus availability
and barley production. Still, the effects of biochar application on the properties of alkaline soils
with high pH and $CaCO_3$ values are not well understood and have received much less
attention.

Despite of positive impacts of biochar on soil quality, the performance of biochar is not
always consistent. For instance, Van Zwieten [22] reported that paper mill waste derived bio-
char increased the production of soybean, while reduced the production of wheat. Likewise,
Jones [23] stated that hardwood derived biochar application significantly improved the soil
fertility, while did not enhance the production of maize. Moreover, the performance of biochar
varies with the characteristics of biochar, which are influenced by feedstock type, pyrolysis
temperature, and resident time. For instance, biochar produced at lower temperature
improved the cation exchange capacity and nutrient availability, whereas, decreased the salin-
ity of the soil. Likewise, biochar produced above 500˚C enhanced soil pH and decreased soil
metal availability [24]. Therefore, selection of a biochar with distinct structural and chemical
properties is a key factor in order to improve soil fertility and productivity.

The selection of suitable and low-cost feedstock for biochar production is of critical impor-
tance in order to obtain maximum benefits. Recycling solid waste materials could serve as a
potential low-cost strategy for biochar production. For instance, re-using solid olive-derived
waste might serve as a potential low-cost technology to produce biochar for soil application.
The amount of olive mill solid waste (OMSW) produced globally accounted for $4 \times 10^8$ kg dry
matter per year, comprising of 38–50% (w/w) cellulose, 23–32% hemi-cellulose, and 15–25%
lignin. Therefore, converting OMSW into biochar could reduce surface pollution on one

hand, and the produced biochar could serve as a soil conditioner for enhanced soil productivity on the other hand. Previous studies have demonstrated the enhancing effects of OMSW-derived biochar on soil microbial biomass C and N, and alteration of the structure of the bacterial community in soil [25, 26]. However, to date, there is no available information on the effects of OMSW-derived biochar on nutrient availability in alkaline sandy soils. Therefore, we investigated (i) the effects of pyrolysis temperature (300–700˚C) on the properties of OMSW-derived biochar, and (ii) potential negative versus positive effects of OMSW-derived biochar on the chemical properties and nutrients availability in calcareous loamy sand soil and on maize (*Zea mays* L.) growth.

## Material and methods

### Production and characteristics of biochar

The solid waste from the olive mill was collected from the Al-Jawf region, Saudi Arabia, and there is no specific permission was required from the company of olive presses to collect the OMSW samples. The OMSW feedstock was dried at 60˚C and then pyrolyzed in a closed container by furnaces under limited oxygen supply at temperatures 300, 400, 500, 600, and 700˚C for 4 h at 5˚C min$^{-1}$. Biochar samples were collected, cooled in a desiccator, ground, sieved through a 2 mm sieve, labelled as BC300, BC400, BC500, BC600, or BC700, and stored for further analyses. The measurements of olive mill solid waste-derived biochar (OMSW-BCs) samples were carried out in duplicate.

In OMSW-derived BC, the moisture content, and that of volatile and resident materials, and ash (proximate characteristics) were analyzed by the ASTM D1762-84 method [27]. The pH of OMSW-derived BCs was measured in a mixture (1:25, w/v) of BC and deionized water using digital pH meter. After measuring biochar pH, the mixture of BC and water was extracted to determine the electrical conductivity (EC) using a digital EC meter. Zeta potential of the BC samples was determined by dynamic light scattering techniques, by measuring the electrophoretic mobility of 1 g L$^{-1}$ BC suspensions in deionized water (Zetasizer Nano ZS, Malvern, UK). Additionally, BC samples were analyzed with a scanning electron microscope (SEM; FEI, Inspect S50), X-ray diffraction (XRD; XRD-7000; Shimadzu Corp, Kyoto, Japan), surface area analyzer (ASAP 2020, Micromeritics, USA), and the Fourier transformation infrared method (Nicolet 6700 FTIR).

### Greenhouse pot experiment

The soil samples were collected from an agricultural farm (24$^o$21'33.1' N and 47$^o$07'49.8' E, altitude 467 m), which is located in a dry land region at Al-Kharaj, Riyadh, Saudi Arabia. No specific permission was required from the farm owner to collect the composite soil samples. The collected composite soil samples were air-dried, sieved, and analyzed for their physicochemical properties. To identify particle size distribution according to Bouyoucos [28], the hydrometer method was applied. According to Sparks [29], and Nelson and Sommers [30], chemical soil properties, including pH, EC, CaCO$_3$, and organic matter were measured. The data for soil analyses showed that the soil samples, which were characterized as having a loamy sand texture (containing 80.89% sand, 12.07% silt, and 7.04% clay) and a low level of organic matter (0.12%), had an alkaline pH (measured in 1:1 suspension of soil to water) value of 7.8, EC (measured in 1:1 extracts of soil to water) value of 2.0 dS m$^{-1}$, and high CaCO$_3$ content (16.51%).

A greenhouse pot experiment with maize plants (*Zea mays* L.) was conducted. Specifically, 1 kg of soil treated with 1% and 3% (w/w) of unwashed OMSW-derived BC at various pyrolytic temperatures (300˚C, BC300; 500˚C, BC500; and 700˚C, BC700). Additionally, treatments

consisting of a control (CK) and inorganic fertilizer of NPK (INF) were applied in the study for comparison. Then, the treated and untreated soil samples were placed in pots, irrigated at the level of field capacity, and incubated for 21 d under laboratory conditions (at a temperature of $23 \pm 2 ^\circ C$). After the incubation period, three replications of the treated and untreated pots were transferred to the greenhouse and placed in a randomized complete block design. Ten maize seeds were sown in each pot. After seedlings emerged, the plants in each pot were thinned to four. The planting period lasted 4 weeks. During the growth period, the plants were irrigated and maintained at 70% of field capacity by weight loss. After 4 weeks of cultivation, the shoots of maize plants and soil were collected from the pots. The collected soil samples were air-dried and analyzed for pH, EC, AB-DTPA-extractable nutrients (P, Fe, Mn, Zn, and Cu), and ammonium acetate-($NH_4OAc$)-extractable K, Na, Ca, and Mg [29]. Additionally, the shoots of maize plants were collected, dried at $70 ^\circ C$, and analyzed for P, Ca, Mg, K, Na, Fe, Mn, Zn, and Cu. The maize plant shoot samples were dried and digested using a dry ashing method [31]. In this study, inductively coupled plasma (ICP, Perkin Elmer Optima 4300 DV ICP-OES) was used to measure the concentrations of P, Ca, Mg, Fe, Mn, Zn, and Cu. The concentrations of K and Na were analyzed using a flame photometer.

## Statistics

For the properties of biochar samples, the means and standard deviations (±SD) are computed. Moreover, to compare the effects of biochar treatments on soil and plant, the differences of means were analysed statistically by a one-way analysis of variance (ANOVA) using the TIBCO Statistica 13.5.0 software [32]. In addition, the obtained data were evaluated by using the least significant difference (LSD) test for post hoc comparisons (at the 0.05 level of significance).

## Results and discussions

### Characteristics of OMSW-derived BC

The results showed that BC yield declined with increasing thermal decomposition during the pyrolytic process (Table 1). The obtained yield of biochar produced at $300 ^\circ C$ accounted for 40.3%, whereas it declined by 32.1%, 28.7%, 27.3%, and 26.7% as the pyrolysis temperature increased to 400, 500, 600, and $700 ^\circ C$, respectively. This reduction was mainly caused by the organic material decomposition and the dehydration of OH groups during the pyrolysis process. However, increasing thermal decomposition during the pyrolytic process increased the levels of fixed carbon and ash in the OMSW-derived BC (Table 1). Several other researchers have reported that the pyrolysis temperature is considered one of the main factors determining BC properties, and thus, the effects of its application in the environment [33–35]. In their studies, the levels of fixed carbon, ash, and pH increased with increasing temperature of the pyrolytic process, whereas yield and volatile matter of BCs tended to decrease. They suggested that high pyrolysis-temperature BCs possess more carbonaceous materials. Additionally, pyrolysis temperature could affect the thermal stability and chemistry of the BC surface [35].

The FTIR results showed that OMSW-derived BC300 possessed a spectrum at 3430 cm$^{-1}$ (S1 Fig), generally attributed to O–H. The spectra of O–H declined with pyrolysis temperature. Additionally, OMSW-derived BC300 exhibited broad bands in the region of 2850–2919 cm$^{-1}$, mainly because of the C-H stretch caused by aliphatic compounds, waxes, and fatty acids. Additionally, the band at 2850 was attributed to symmetrical CH in–$CH_2$, suggesting the presence of fatty acids and alkanes. These two peaks at 2850 and 2919 cm$^{-1}$ declined in BC400 and completely disappeared with a temperature $\geq 500 ^\circ C$. The absorption at 1574–1652 cm$^{-1}$ of biochar samples suggested the presence of–COOH, as well as C = O and C = C, especially in

**Table 1. pH, Electrical Conductivity (EC), and approximate analysis of Olive Mill Solid Waste-derived Biochar (OMSW-BCs) samples as affected by pyrolysis temperature.**

| Parameters | Samples[1] | | | | | |
|---|---|---|---|---|---|---|
| | **FS** | **BC300** | **BC400** | **BC500** | **BC600** | **BC700** |
| pH | $5.97 \pm 0.02$[2] | $9.85 \pm 0.08$ | $10.12 \pm 0.02$ | $10.03 \pm 0.02$ | $10.11 \pm 0.03$ | $10.21 \pm 0.04$ |
| EC, (dS m$^{-1}$) | $0.83 \pm 0.01$ | $1.06 \pm 0.02$ | $2.11 \pm 0.02$ | $2.09 \pm 0.02$ | $2.39 \pm 0.00$ | $2.54 \pm 0.04$ |
| Yield, % | - | $40.3 \pm 0.47$ | $32.1 \pm 0.12$ | $28.7 \pm 0.82$ | $27.3 \pm 0.03$ | $26.7 \pm 0.49$ |
| Moisture, % | $6.11 \pm 0.49$ | $1.16 \pm 0.006$ | $1.44 \pm 0.007$ | $1.72 \pm 0.03$ | $1.89 \pm 0.02$ | $0.95 \pm 0.02$ |
| Volatile matter, % | $74.51 \pm 1.04$ | $27.74 \pm 1.33$ | $17.07 \pm 0.49$ | $11.96 \pm 0.44$ | $11.00 \pm 0.02$ | $9.62 \pm 0.02$ |
| Ash, % | $2.02 \pm 0.08$ | $6.91 \pm 0.15$ | $8.92 \pm 0.18$ | $9.43 \pm 0.30$ | $9.59 \pm 0.17$ | $9.62 \pm 0.03$ |
| Fixed C, % | $17.36 \pm 1.44$ | $64.19 \pm 1.19$ | $72.57 \pm 0.60$ | $76.90 \pm 0.11$ | $77.51 \pm 0.16$ | $79.80 \pm 0.03$ |

1. FS: feedstock; BC300: biochar produced at 300°C; BC400: biochar produced at 400°C; BC500: biochar produced at 500°C; BC600: biochar produced at 600°C; BC700: biochar produced at 700°C.

2. Values represent the means ± standard deviations (±SD).

an aromatic form. However, the intensity of these peaks (1574–1652 cm$^{-1}$) decreased in BC samples produced at higher pyrolysis temperatures ($\geq$ 500°C). The absorption at approximately 1400 cm$^{-1}$ (with an intense band for BC300 and BC400) suggested the presence of aliphatic and aromatic O-H groups, which decreased at higher pyrolysis temperatures.

In this study, XRD was used to identify the mineral composition of BC samples (S2 Fig). In the samples of OMSW-derived BC300, peaks at a spacing of 4.02, 2.06, 1.45, and 1.23 Å were identified, suggesting the presence of kalcinite [K(HCO$_3$)], sylvite (KCl), and perclase (MgO), which were reduced or disappeared with increasing pyrolysis temperatures. Furthermore, a high pyrolysis temperature resulted in calcite minerals (CaCO$_3$) at 3.04–3.14, 1.87, 1.81 Å. Additionally, the SEM analysis depicted a greater change in the surface structure of BCs compared to that of feedstock with temperature (S3 Fig).

Larger zeta potentials were observed at the lower pyrolysis temperatures of 300 and 400°C (Table 2), which was mainly attributed, as indicated by the FTIR results, to the increased number of O–H and–COOH groups observed in the BC produced at these temperatures. As a result of the decline or the complete disappearance of the O–H and–COOH groups at higher pyrolysis temperatures, less negative charges remained on the surfaces of BC, and zeta

**Table 2. Zeta potential, surface area and pore properties of Olive Mill Solid Waste-derived Biochars (OMSW-BCs).**

| Pyrolysis temperature (°C) | Zeta potential (mV) | SBET[2] (m$^2$ g$^{-1}$) | S$_{micr}$[3] (m$^2$ g$^{-1}$) | Vt[4] (cm$^3$ g$^{-1}$) | V$_{micro}$[5] (cm$^3$ g$^{-1}$) | D$_{ave}$[6] (nm) |
|---|---|---|---|---|---|---|
| 300 | $-43.0 \pm 1.64$[1] | $0.35 \pm 0.01$ | $0.35 \pm 0.01$ | $0.0015 \pm 0.00$[7] | $0.00018 \pm 0.00$ | $170.8 \pm 6.4$ |
| 400 | $-50.2 \pm 1.96$ | $1.78 \pm 0.04$ | $1.78 \pm 0.03$ | $0.0021 \pm 0.00$ | $0.00086 \pm 0.00$ | $46.1 \pm 1.7$ |
| 500 | $-29.3 \pm 1.21$ | $108.4 \pm 1.9$ | $80.5 \pm 1.6$ | $0.055 \pm 0.00$ | $0.037 \pm 0.00$ | $20.4 \pm 0.7$ |
| 600 | $-27.3 \pm 1.53$ | $128.0 \pm 2.3$ | $109.2 \pm 2.2$ | $0.063 \pm 0.00$ | $0.051 \pm 0.00$ | $19.84 \pm 0.9$ |
| 700 | $-24.7 \pm 0.75$ | $168.4 \pm 3.2$ | $143.8 \pm 3.1$ | $0.083 \pm 0.00$ | $0.066 \pm 0.00$ | $19.74 \pm 0.6$ |

1. Errors are ± SD.

2. SBET surface area.

3. Microporous surface area by the t-plot method.

4. Total pore volume

5. Microporous pore volume by the t-plot method.

6. Average pore width, estimated by 4Vt/SBET.

7. ±SD reported as zero have values $\leq 1.0 \times 10^{-5}$

potential were decreased for BCs produced at 500, 600, and 700˚C. The surface area and pore characteristics of OMSW-derived BCs showed an improvement in the surface area and total and microporous pore volumes with pyrolysis temperature (Table 2). OMSW-derived BCs made at the lowest temperatures of 300˚C and 400˚C, respectively, possessed the lowest surface area of 0.35 m$^2$ g$^{-1}$ and 1.78 m$^2$ g$^{-1}$; however, OMSW-derived BC500, BC600, and BC700 had high values for the surface area of 108.4, 128.0, and 168.4 m$^2$ g$^{-1}$, respectively. In contrast, as pyrolysis temperature increased from 300˚C to 400˚C, 500˚C, 600˚C, and 700˚C, the average pore width for OMSW-derived BCs decreased from to 170.8 to 46.1, 20.4, 19.8, and 19.7 nm, respectively. Generally, during the pyrolytic process, the loss of volatile materials and the functional groups H- and O- were the main reason for the improvement in surface area and pore characteristics of the resultant BCs. In this context, pyrolytic temperatures were positively correlated with BC characteristics, including EC (r = 0.885), fixed carbon (r = 0.9264), surface area (r = 0.9562), microporous surface area (r = 0.9691), total pore volume (r = 0.9532), and microporous pore volume (r = 0.9690) (Table 3). However, pyrolysis temperature showed a negative correlation with yield (r = -0.9054) and volatile matter (r = -9028). In general, the greatest changes were pronounced for high pyrolysis temperatures ≥500˚C. When plotting the biochar characteristics (yield, volatile matter, fixed carbon, ash content, and surface area) in relation to different pyrolysis temperatures, the best model fitting was pronounced with a second-order polynomial (S4 Fig).

## Biochar effects on soil pH and EC

OMSW-derived BC treatments increased soil pH from 7.85 to 8.05, 8.15, and 8.11 at an application rate of 1% for BC300, BC500 and BC700, respectively (Table 4). Meanwhile, at a high application rate of 3%, these corresponding values of soil pH increased to 8.28 (BC300), 8.31 (BC500) and 8.32 (BC700). The occurred increases in soil pH, due to BC application, are mainly because of the alkalinity induced by BCs as a result of the basic oxides and carbonates produced during the pyrolytic process. Previous studies have also indicated the application effects of BCs on increasing soil pH [36–38]. However, in contrast to our results, other studies have found that alkaline soil pH decreased with biochar addition [14]. The reductions in soil pH could be attributed to BC oxidation and the production of acidic materials. In the current study, the higher pH of produced BCs (9.85–10.21) compared to the control soil (7.80) could result in increasing soil pH. The increases in soil pH could have occurred because of the increase in the level of soil exchangeable base cations with BC addition. In a study conducted on calcareous soil, Cardelli et al. [36] suggested that the alkalizing effects of BC on soil pH could be explained by the poor buffering capacity of soil induced by the low levels of soil organic matter.

In this study, the EC values for control soils were 2.03 dS m$^{-1}$ (CK) and 2.08 dS m$^{-1}$ (CK +NPK) (Table 4). These EC values increased to 2.15 dS m$^{-1}$ for 3% BC300, to 2.16 (BC500 at 1%) and 2.21 dS m$^{-1}$ (BC500 at 3%), and to 2.10 (BC700 at 1%) and 2.20 dS m$^{-1}$ (BC700 at 3%). It has been previously reported that a high level of soil salinity following BC application was caused by the introduction of high levels of soluble ions derived from BC ash into the soil [39].

## BC effects on soil nutrient availability

The incorporation of BC into soils can affect soil chemistry, causing changes in the available fraction of soil P, mainly because of alterations in soil pH and cation exchange capacity (CEC). Although soil pH in the current study increased because of the application of OMSW-derived BC, soil AB-DTPA-extractable P concentrations exhibited some increases in the INF and BC treatments compared to that of the control soil, but they were not significant (Table 4). For

**Table 3. Pearson correlation coefficient (r) among Olive Mill Solid Waste-derived Biochar (OMSW-BCs) properties.**

| BC properties | Pyrolysis temperature | pH | EC | Zeta potential | Yield | Volatile matter | Ash | Fixed carbon | SBET | $S_{mic}$ | $V_t$ | $V_{mic}$ | $D_{ave}$ |
|---|---|---|---|---|---|---|---|---|---|---|---|---|---|
| Pyrolysis temperature | 1.0000 | | | | | | | | | | | | |
| pH | 0.8280 | 1.0000 | | | | | | | | | | | |
| EC | 0.8850* | 0.9590 | 1.0000 | | | | | | | | | | |
| Zeta potential | -0.8475 | -0.4195 | -0.5800 | 1.0000 | | | | | | | | | |
| Yield | -0.9045* | -0.8730 | -0.9726* | 0.7127 | 1.0000 | | | | | | | | |
| Volatile matter | -0.9028* | -0.8652 | -0.9662* | 0.7208 | 0.9990* | 1.0000 | | | | | | | |
| Ash | 0.8416 | 0.8797 | 0.9738* | -0.6175 | -0.9903* | -0.9891* | 1.0000 | | | | | | |
| Fixed carbon | 0.9264* | 0.8751 | 0.9646* | -0.7422 | -0.9947* | -0.9970* | 0.9779* | 1.0000 | | | | | |
| SBET | 0.9562* | 0.6468 | 0.7627 | -0.9625* | -0.8476 | -0.8534 | 0.7683 | 0.8771 | 1.0000 | | | | |
| $S_{mic}$ | 0.9691* | 0.6680 | 0.7693 | -0.9513* | -0.8398 | -0.8429 | 0.7570 | 0.8687 | 0.9966* | 1.0000 | | | |
| $V_t$ | 0.9532* | 0.6426 | 0.7604 | -0.9637* | -0.8475 | -0.8540 | 0.7688 | 0.8776 | 0.9999* | 0.9953* | 1.0000 | | |
| $V_{mic}$ | 0.9690* | 0.6673 | 0.7702 | -0.9517* | -0.8409 | -0.8435 | 0.7581 | 0.8686 | 0.9963* | 0.9999* | 0.9949* | 1.0000 | |
| $D_{ave}$ | -0.7926 | -0.8689 | -0.9611* | 0.5578 | 0.9745* | 0.9742* | -0.9960* | -0.9590* | -0.7147 | -0.6997 | -0.7159 | -0.7006 | 1.0000 |

SBET: surface area; $S_{mic}$: Microporous surface area; $V_t$: Total pore volume; $V_{mic}$: Microporous pore volume; $D_{ave}$: Average pore width

example, inorganic fertilizer treatment increased soil available P by 1.28 mg kg$^{-1}$, whereas the BC treatments increased its availability by 0.05–1.13 mg kg$^{-1}$. In a meta-analysis study conducted by Glaser and Lehr [17] on the availability of soil P, the addition of BCs to acidic and neutral soils significantly enhanced the availability of soil P, whereas there was no significant effect in alkaline soils. Thus, contrary to the effects in acidic soils, it would be advisable to

**Table 4. The influences of the applied Olive Mill Solid Waste-derived Biochars (OMSW-BCs) on the soil pH and EC, and the soil concentrations of NH$_4$OAc- and Ammonium Bicarbonate-DiethyleneTriaminePentaacetic Acid (AB-DTPA)-extractable nutrients.**

| Treatments[1] | pH | EC | NH$_4$OAc-extractable nutrients | | | | AB-DTPA-extractable nutrients | | | | |
|---|---|---|---|---|---|---|---|---|---|---|---|
| | | | K | Na | Ca | Mg | P | Fe | Mn | Cu | Zn |
| | | dS m$^{-1}$ | ·····································································  mg kg$^{-1}$ ·································································· | | | | | | | | |
| CK | 7.85a[2] | 2.03a | 40.5a | 26.0ab | 8,140a | 105a | 10.09a | 0.07a | 0.66a | UDL[3] | 0.31ab |
| INF | 7.88a | 2.08ab | 65.0b | 26.0ab | 8,080a | 99.3a | 11.37a | 0.07a | 0.46bc | UDL | 0.23a |
| BC300-1% | 8.05b | 2.04a | 102c | 24.8ac | 8,077a | 94.8a | 10.83a | 0.09a | 0.57ab | UDL | 0.30ab |
| BC300-3% | 8.28c | 2.15cd | 209d | 27.8bd | 8,090a | 93.9a | 10.92a | 0.06a | 0.66a | UDL | 0.34ab |
| BC500-1% | 8.15d | 2.16cd | 130e | 27.0bd | 8,005a | 98.4a | 10.51a | 0.06a | 0.40c | UDL | 0.35bc |
| BC500-3% | 8.31c | 2.21e | 324f | 32.5e | 7,943a | 91.1a | 10.14a | 0.05a | 0.49bc | UDL | 0.27ab |
| BC700-1% | 8.11bd | 2.10b | 135e | 26.8ab | 8,003a | 99.5a | 10.73a | 0.17b | 0.42c | UDL | 0.28ab |
| BC700-3% | 8.32c | 2.20cde | 360j | 30.8e | 9,962b | 134b | 11.22a | 0.22b | 0.51bc | UDL | 0.45c |
| Analysis of variance (ANOVA) | | | | | | | | | | | |
| F-value | 44.646 | 18.448 | 333 | 7.719 | 1.774 | 2.601 | 1.038 | 6.717 | 6.841 | - | 3.225 |
| P-value | 0.0000 | 0.0000 | 0.0000 | 0.0003 | 0.1620 | 0.0537 | 0.4434 | 0.0008 | 0.0007 | - | 0.0248 |

1. CK: control; INF: inorganic fertilizer (NPK); BC300-1%: biochar produced at 300˚C and added at 1% application rate; BC300-3%: biochar produced at 300˚C and added at 3% application rate; BC500-1%: biochar produced at 500˚C and added at 1% application rate; BC500-3%: biochar produced at 500˚C and added at 3% application rate; BC700-1%: biochar produced at 700˚C and added at 1% application rate; BC700-3%: biochar produced at 700˚C and added at 3% application rate.

2. Different letters indicate significant differences according to the LSD test (p = 0.05, n = 3).

3. UDL: under detection limit.

apply acidic biochar (not alkaline biochar) to alkaline soils with P constraints to improve the levels of plant-available P.

$NH_4OAc$-extractable K increased significantly by BC application to the soil (Table 4). In INF-treated soil, the soil exchangeable content of K significantly increased from 40.5 to 65.0 mg kg$^{-1}$. However, because of the application of BCs, the values of soil exchangeable K content greatly increased from 40.5 (CK) to 102 mg kg$^{-1}$ (1% BC300), 209 mg kg$^{-1}$ (3% BC300), 130 mg kg$^{-1}$ (1% BC500), 324 mg kg$^{-1}$ (3% BC500), 135 mg kg$^{-1}$ (1% BC700), and 360 mg kg$^{-1}$ (3% BC700). Notably, significant differences among BC treatments were found in the soil content of exchangeable K. Therefore, both pyrolysis temperature and application rates of BCs had a significant effect on the increase in the available form of soil K. Additionally, application of high rates of BCs pyrolyzed at high temperature (500°C and 700°C) exhibited greater significant increases in soil exchangeable Na than that in the control soil (Table 4), and exhibited increases from 26.0 mg kg$^{-1}$ (CK) to 32.5 mg kg$^{-1}$ (3% BC500) and 30.8 mg kg$^{-1}$ (3% BC700). The experimental results of Jien and Wang [40] indicated that the levels of exchangeable K were significantly enhanced in BC-treated soil, suggesting that it improved the level in the soil. Our results suggested that OMSW-derived BC itself might be a K source, and thus, it enhanced its bioavailability in soils. It has also been reported that BC application could increase the concentrations of exchangeable Ca and Mg in the soil [40]. In the present study, among all BC treatments, only 3% BC700 resulted in significant increases in the concentrations of $NH_4OAc$-extractable Ca and Mg by 22.4% and 27.9% compared to that of the control soil, respectively. This suggests that the availability of soil Ca and Mg in OMSW-derived BC could likely depend on both pyrolysis temperature and application rate. The high content of soil exchangeable basic cations in the high pyrolysis-temperature BC-treated soil could be explained by the increasing content of BC ash. Several other researchers have explained the improvements in the exchangeability of soil cations because of the presence of ashes in BC, which contain high levels of oxides and hydroxides of alkali cations [41].

Regarding micronutrient availability in soil, BC significantly affected the soil concentrations of AB-DTPA-extractable micronutrients, depending upon pyrolysis temperature and application rates (Table 4). The concentrations of AB-DTPA-extractable Mn in INF, BC500, and BC700 treatments were lower than that in the CK. Although BC treatments caused significant increases in soil pH, application of BC pyrolyzed at the highest temperature (BC700) resulted in a significant improvement in the soil concentrations of AB-DTPA-extractable Fe (at application rates of 1% and 3%) and Zn (at an application rate of 3%). The soil concentrations of AB-DTPA-extractable Fe increased significantly from 0.07 mg kg$^{-1}$ in the CK to 0.17 mg kg$^{-1}$ with 1% BC700 and 0.22 mg kg$^{-1}$ with 3% BC700, whereas AB-DTPA-extractable Zn increased significantly from 0.31 mg kg$^{-1}$ in the CK to 0.45 mg kg$^{-1}$ with 3% BC700. Among all micronutrients, soil concentrations of AB-DTPA-extractable Cu were undetectable by ICP-OES.

## BC effects on dry matter and mineral content of maize plants

Statistically, BC addition did not significantly affect plant growth parameters (fresh and dry weight and plant height) (Table 5). Similarly, Farrell et al. [42] found no improvement in wheat yield in calcareous soils because of BC addition. They suggested that the addition of BC to calcareous soil did not lead to better conditions for the uptake of plant nutrients. In this context, the results of the current study showed that the shoots of maize plants in BC-amended soils had significantly lower content of P, Ca, and Mg, especially with increasing application rate. In soil treated with high application rate (3%) of OMSW-derived BC, the shoot P content decreased from 2.24 g kg$^{-1}$ (CK) to 1.68 g kg$^{-1}$ (BC300), 1.82 g kg$^{-1}$ (BC500), and 1.88 g kg$^{-1}$

**Table 5. Olive Mill Solid Waste-derived Biochar (OMSW-BCs) effects on plant height, fresh and dry matter, and shoot mineral content of *Zea mays*.**

| Treatments[1] | Plant height (cm) | Plant fresh weight (g plant⁻¹) | Plant dry weight (g plant⁻¹) | K | Na | P | Ca | Mg | Fe | Mn | Zn | Cu |
|---|---|---|---|---|---|---|---|---|---|---|---|---|
| | | | | g kg⁻¹ | | | | | mg kg⁻¹ | | | |
| CK | 54.5ab[2] | 2.37a | 0.21a | 15.1a | 0.14a | 2.24a | 13.06ab | 10.6a | 108a | 49.5ab | 64.5a | 8.94a |
| INF | 54.3ab | 2.56a | 0.22a | 23.4ab | 0.16ab | 2.41ab | 16.60c | 9.51b | 106a | 53.9a | 76.3ab | 8.22a |
| BC300-1% | 54.2ab | 2.59a | 0.20a | 33.2bc | 0.26ac | 1.71c | 12.37bd | 6.45c | 91.3a | 43.1cd | 74.3ab | 7.39a |
| BC300-3% | 50.2b | 2.58a | 0.18a | 56.1de | 0.30c | 1.68c | 10.50d | 5.08d | 105a | 39.4d | 71.9ab | 9.67a |
| BC500-1% | 53.6ab | 2.76a | 0.21a | 49.9df | 0.28bc | 2.18ab | 11.61bd | 5.90ce | 105a | 47.1bc | 81.8b | 8.11a |
| BC500-3% | 49.7b | 2.53a | 0.20a | 65.3ej | 0.31c | 1.82c | 7.64e | 3.88f | 92.4a | 43.1cd | 79.3b | 7.56a |
| BC700-1% | 59.4a | 3.17a | 0.24a | 39.4cf | 0.51d | 2.17ab | 11.35bd | 6.33c | 108a | 46.2bc | 77.4b | 7.72a |
| BC700-3% | 49.4b | 2.62a | 0.21a | 73.7j | 0.79e | 1.88c | 6.23e | 3.71f | 97.7a | 40.2d | 69.2ab | 7.22a |
| Analysis of variance (ANOVA) | | | | | | | | | | | | |
| *F-value* | 1.483 | 0.7809 | 0.3612 | 31.258 | 23.692 | 10.113 | 24.560 | 96.687 | 0.8794 | 7.672 | 1.790 | 0.8221 |
| *P-value* | 0.2421 | 0.6126 | 0.9117 | 0.0000 | 0.0000 | 0.0001 | 0.0000 | 0.0000 | 0.5436 | 0.0004 | 0.1584 | 0.5832 |

1. CK: control; INF: inorganic fertilizer (NPK); BC300-1%: biochar produced at 300˚C and added at 1% application rate; BC300-3%: biochar produced at 300˚C and added at 3% application rate; BC500-1%: biochar produced at 500˚C and added at 1% application rate; BC500-3%: biochar produced at 500˚C and added at 3% application rate; BC700-1%: biochar produced at 700˚C and added at 1% application rate; BC700-3%: biochar produced at 700˚C and added at 3% application rate.

2. Different letters indicate significant differences according to the LSD test (p = 0.05, n = 3).

(BC700). The shoot content of Ca decreased from 13.06 g kg⁻¹ in the CK to 10.50 g kg⁻¹, 7.64 g kg⁻¹, and 6.23 g kg⁻¹ for BC300, BC500, and BC700 at 3% application rate, respectively. Furthermore, the shoot content of Mg decreased from 10.6 g kg⁻¹ in the CK to 5.08, 3.88, and 3.71 g kg⁻¹ for BC300, BC500, and BC700 at 3% application rate, respectively. In alkaline soils, calcareous substances can lead to the formation of insoluble compounds of Ca-Mg-P, decreasing the shoot content of these nutrients [43]. On the other hand, the great quantity of K obtained because of OMSW-derived BC could result in reduced plant uptake of Ca, Mg, and P by an antagonistic effect [44, 45]. Contrary to our results, several reports found increases in the plant content of nutrients [41, 46]. The discrepancy between our data and that of other studies could be explained because of varying soil characteristics and feedstock used to produce BCs. Our results suggested that the incorporation of OMSW-derived BCs into alkaline soils may limit the essential nutrient (such as Ca, Mg, and P) uptake by plants, and require the additional input of these nutrients (especially P fertilizer). Further research on OMSW-derived BCs is needed to clarify the mechanism responsible for governing uptake of these nutrients by plants.

Contrary to that of shoot P, Ca, and Mg, OMSW-derived BCs treatments enhanced the levels of K and Na in plant shoots (Table 5). In soil treated with 1% and 3% OMSW-derived BC, respectively, the shoot K content increased from 15.1 g kg⁻¹ (CK) and 23.4 g kg⁻¹ (INF) to 33.2 and 56.1 g kg⁻¹ (BC300), 49.9 and 65.3 g kg⁻¹ (BC500), and 39.4 and 73.7 g kg⁻¹ (BC700). This indicated that shoot K content increased with increasing pyrolysis temperature and application rate. In this study, soil exchangeable K was also significantly increased following BC application and it increased with both increasing pyrolysis temperature and application rate, reflecting enhanced K uptake by plant shoots. It has been reported that a high quantity of K in BCs could enhance the bioavailable pool of K in the soil [47]. Application of BCs might also improve soil mineral K release by enhancing the activity of K-dissolving bacteria and facilitating K uptake by crops [48]. Syuhada et al. [49] found that the K content in corn tissue was significantly higher but its tissue content of N, Ca, and Mg was lower in BC-amended plants than that of the control. They attributed the high tissue K content in BC treatments to increasing exchangeable soil K.

BCs exhibited varying effects on the shoot content of micronutrients, depending upon the type of micronutrient. Statistically, BC addition did not have a significant effect on the shoot levels of Fe and Cu (Table 5). However, the addition of BCs (especially at the highest application rate) significantly decreased the shoot Mn levels in comparison with that of the CK. Conversely, application of 1% and 3% BC500 and 1% BC700 significantly enhanced the level of shoot Zn.

### Recommendations and suggestions for future trials

This study was conducted to evaluate the effects of application of OMSW-derived BCs (in relation to pyrolysis temperature and application rates) on soil pH, EC, availability of soil nutrients to plants, and maize growth in arid alkaline soil. The results showed that the properties of BC were affected by increasing pyrolytic temperatures, reflecting on soil pH, EC, and the performance of soil nutrients availability to plants. The application of OMSW-derived BC decreased the levels of Ca, Mg, Mn, and P in plant shoots and enhanced the levels of shoots K, Na, and Zn. The high K content in BC treatments suggests high agronomic value in terms of replacing conventional K sources. However, despite the high soil available concentrations of K, the significant increase in soil pH values and decrease in P, Ca, Mg, and Mn of plant tissues because of the application of BC should not be ignored as these nutrients are of substantial importance in terms of agronomic potential. The quantity of K incorporated by OMSW-derived BC could negatively reflect the decreasing nutrient uptake by plants, as indicated in this study for the decrease in shoot content of P, Ca, Mg, and Mn. For OMSW-derived BC, lowering of shoot nutrient content may likely limit its use, and thus, care should be taken in its use under arid conditions. Collectively, our findings suggested that application of OMSW-derived BCs may not be able to provide sufficient nutrients to enhance plant growth; however, because of high alkalinity, OMSW-derived BC may be applied as a soil additive to acidic soils rather than alkaline ones. In the current study, OMSW-derived BCs were applied to a calcareous soil in unwashed form, which may have an opposite impact on the performance of soil properties and nutrients availability. Therefore, further studies on OMSW-derived BCs in washed form are needed to evaluate its effects on a wide range of alkaline and calcareous soils with varied properties in comparison with unwashed OMSW-derived BCs. Further studies are also required on the effects of OMSW-derived BC in combination with inorganic fertilizers on the antagonistic effect induced by the high quantity of K and the properties and fertility of arid-land soils.

### Supporting information

**S1 Fig. FTIR spectra of the produced Olive Mill Solid Waste-derived Biochar (OMSW-BCs) (BC300: biochar produced at 300˚C; BC400: biochar produced at 400˚C; BC500: biochar produced at 500˚C; BC600: biochar produced at 600˚C; BC700: biochar produced at 700˚C).**
(DOCX)

**S2 Fig. XRD spectra of the produced Olive Mill Solid Waste-derived Biochar (OMSW-BCs) (BC300: biochar produced at 300˚C; BC400: biochar produced at 400˚C; BC500: biochar produced at 500˚C; BC600: biochar produced at 600˚C; BC700: biochar produced at 700˚C).**
(DOCX)

**S3 Fig. Scanning electron microscope (SEM) analyses of feedstock (FS) and olive mill solid waste-derived biochars (OMSW-BCs) pyrolyzed at different temperatures (a: FS:**

**feedstock; b: BC300: biochar produced at 300˚C; c: BC400: biochar produced at 400˚C; d: BC500: biochar produced at 500˚C; e: BC600: biochar produced at 600˚C; f: BC700: biochar produced at 700˚C).**
(DOCX)

**S4 Fig. Model fitting for the produced Olive Mill Solid Waste-derived Biochar (OMSW-BCs) properties in relation with pyrolysis temperatures.**
(DOCX)

## Acknowledgments

The authors extend their appreciation to the Deanship of Scientific Research, King Saud University, for funding this work through the international research group project RG-1439-043.

## Author Contributions

**Conceptualization:** Adel R. A. Usman, Mohammad I. Al-Wabel.

**Data curation:** Adel R. A. Usman, Munir Ahmad, Abdelazeem S. Sallam.

**Formal analysis:** Azzaz Alazzaz, Jamal Elfaki, Abdelazeem S. Sallam, Mutair A. Akanji.

**Funding acquisition:** Mohammad I. Al-Wabel.

**Investigation:** Azzaz Alazzaz, Hesham M. Ibrahim, Jamal Elfaki, Mutair A. Akanji.

**Methodology:** Azzaz Alazzaz, Mutair A. Akanji.

**Project administration:** Mohammad I. Al-Wabel.

**Resources:** Mohammad I. Al-Wabel.

**Supervision:** Adel R. A. Usman, Mohammad I. Al-Wabel.

**Validation:** Azzaz Alazzaz, Adel R. A. Usman.

**Visualization:** Azzaz Alazzaz, Adel R. A. Usman, Mohammad I. Al-Wabel.

**Writing – original draft:** Azzaz Alazzaz, Adel R. A. Usman, Munir Ahmad.

**Writing – review & editing:** Azzaz Alazzaz, Adel R. A. Usman, Munir Ahmad, Hesham M. Ibrahim, Mohammad I. Al-Wabel.

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
