## [Decision Letter · Decision Letter 0]

1 Jun 2020

PONE-D-20-11578

Potential short-term negative versus positive effects of olive mill-derived biochar on nutrient availability in a calcareous loamy sandy soil

PLOS ONE

Dear Dr. Al-wabel,

Thank you for submitting your manuscript to PLOS ONE. After careful consideration, we feel that it has merit but does not fully meet PLOS ONE’s publication criteria as it currently stands. Therefore, we invite you to submit a revised version of the manuscript that addresses the points raised during the review process.

We look forward to receiving your revised manuscript.

Kind regards,

Andrés Rodríguez-Seijo, PhD

Academic Editor

PLOS ONE

Journal Requirements:

2. In your Methods section, please provide additional information regarding the permits you obtained to collect samples for the present study. Please ensure you have included the full name of the authority that approved the field site access and, if no permits were required, a brief statement explaining why.

3. Please amend the manuscript submission data (via Edit Submission) to include author Hesham Ibrahim.

Reviewers' comments:

Reviewer's Responses to Questions

**Comments to the Author**

1. Is the manuscript technically sound, and do the data support the conclusions?

Reviewer #1: Yes

Reviewer #2: Yes

2. Has the statistical analysis been performed appropriately and rigorously? 

Reviewer #1: Yes

Reviewer #2: No

3. Have the authors made all data underlying the findings in their manuscript fully available?

Reviewer #1: Yes

Reviewer #2: Yes

4. Is the manuscript presented in an intelligible fashion and written in standard English?

Reviewer #1: Yes

Reviewer #2: Yes

5. Review Comments to the Author

Reviewer #1: This manuscript does 2 things: First it chemically characterized biochars produced by the pyrolysis of olive mill waste and second, it compares zea mays growth in soils with biochar (and controls). The title and abstract stress the greenhouse experiment, yet the results and discussion focus on the chemical analysis of biochar. It is overly long, but also seems lacking a lot of details. It is very repetitive. For it to be acceptable for publications the authors need to re-envision the paper and make it much more clear.

The introduction spends a lot of space talking about what biochar can do for the soil in a really general way. There is so many papers out there on this already you only need to reference them briefly and move on. Instead use the introduction to educate people about what specifically you are actually doing- . What do we already know about how different production temperatures effect biochar – its nutrients, mass, etc? Then lay out predictions for how you would expect your materials to act. Similarly, this needs to be done for plants and soil nutrients – there are so many studies (and meta-analyses) that describe how biochar effects these things.

Yes, this paper is in alkaline soils and that is an interesting aspect, but you need to lay out what we know and then how do we expect that to change in alkaline soils.

THEN your results should only focus on those things laid out in the introduction. Did it do what you expected? Why or why not?. For instance there is no need to reiterate your design or anything in the discussion, implications and conclusions. This should be simply laid out in the introduction (which is just a page or two away)

Minor things

I cannot speak to the chemical analysis of the biochar (XRD etc) I assume this was performed well.

The statistics section requires much more detail

Please indicate in the tables which values are different from each other

Recommendations and conclusions are redundant. The paper is not that long and the conclusions are just not needed.

Line 63 – mitigate climate– add change

Lines 72-75 These sentences are contradictory as one says there has not been many studies in alkaline soils and then the next sentence sites three. I would just delete the last sentence

Line 77 – delete “on the other hand” (there is no first hand)

Lines 82-87 should be tacked on to the paragraph above it. The last paragraph of the sentence should start “Previous studies…”

Line 93 - Replace “the aims of this study were to investigate” to “We investigated”. -this gets rid of redundant words and is in an active voice.

Delete line 143-144 – one can just put “Table 1” at the end of the next sentence

All tables need to be formatted similarly and to the journal’s standards.

Line 152- what is basicity (do you mean pH?)

Line 209 –This paragraph is very confusing and overly wordy – please simplify

Reviewer #2: The production of BC from OMSW is an important area of research

In the section of Material and methods it is not clear if the BC was washed before being mixed with the soil, otherwise some of contaminants were introduced into the soil

From the beginning you selected alkaline soil and the BC is going to increase that. Did you try some acidic soil which is more suitable for your BC?? Why not from the beginning you selected to compare acidic vs alkaline soil?? Your BC is better suit acidic soil

The Zeta potential was not measured and data should be supplied, otherwise it is difficult to explain why for example the nutrients such as Ca, Mg and P and others were limited to the plants and the BC was unable to adsorb them

The Tables 1 and 2 no mention how many replicates, SD and statistical analysis such as LSD are available. You are talking about significance, but I can't see that.

I don't see data concerning N content in all soils especially you used NPK in the INF treatment.

You see from the results (for example Table 5) that when you have increased the soil amendment from 1% to 3% you see increase in the negative effects on plant parameters which means you have contamination in your BC which means you should wash it!!!

Might be that your supplied BC have affected your soil properties and plant growth as well. Accordingly, your results are indicating that BC supplementation didn't improve plant growth parameters.

6. PLOS authors have the option to publish the peer review history of their article (what does this mean?). If published, this will include your full peer review and any attached files.

Reviewer #1: No

Reviewer #2: No

---

## [Author Response · Author response to Decision Letter 0]

7 Jun 2020

Response to Reviewers

Dear Editor of PLOS ONE

On behalf of the authors of the manuscript entitled "Potential short-term negative versus positive effects of olive mill-derived biochar on nutrient availability in a calcareous loamy sandy soil", I wish to express my respects for you and for the reviewers for reviewing this manuscript. The authors have modified the manuscript based on the valuable reviewer’s comments and responses to reviewers’ comments are addressed point by point and given below. 

Journal Requirements:

2. In your Methods section, please provide additional information regarding the permits you obtained to collect samples for the present study. Please ensure you have included the full name of the authority that approved the field site access and, if no permits were required, a brief statement explaining why.

3. Please amend the manuscript submission data (via Edit Submission) to include author Hesham Ibrahim.

Response: All above requirements have been done and/or provided. 

Reviewer #1: 

Specific comments:

Comment 1: This manuscript does 2 things: First it chemically characterized biochars produced by the pyrolysis of olive mill waste and second, it compares zea mays growth in soils with biochar (and controls). The title and abstract stress the greenhouse experiment, yet the results and discussion focus on the chemical analysis of biochar. It is overly long, but also seems lacking a lot of details. It is very repetitive. For it to be acceptable for publications the authors need to re-envision the paper and make it much clearer.

Response: We are thankful to the reviewer for the critical comments of the reviewer. Based on your valuable comments, we revised our manuscript. 

Comment 2: The introduction spends a lot of space talking about what biochar can do for the soil in a really general way. There is so many papers out there on this already you only need to reference them briefly and move on. Instead use the introduction to educate people about what specifically you are actually doing- . What do we already know about how different production temperatures effect biochar – its nutrients, mass, etc? Then lay out predictions for how you would expect your materials to act. Similarly, this needs to be done for plants and soil nutrients – there are so many studies (and meta-analyses) that describe how biochar effects these things.

Yes, this paper is in alkaline soils and that is an interesting aspect, but you need to lay out what we know and then how do we expect that to change in alkaline soils. THEN your results should only focus on those things laid out in the introduction. Did it do what you expected? Why or why not?. For instance there is no need to reiterate your design or anything in the discussion, implications and conclusions. This should be simply laid out in the introduction (which is just a page or two away).

Response: We are highly thankful to the reviewer for valuable suggestions. The introduction has been revised extensively in the revised version of the manuscript. More focus has been drawn on the information related to the work performed in this study. The information about the impacts of biochar with different pyrolyzing temperature has been added. More information on expected performance of the biochar in calcareous alkaline soils has been added now. Moreover, the possible negative impacts of biochar are also mentioned as the performance of biochar is not always consistent. Thus, revised version is now possessing improved quality. 

Minor things

Comment 1: I cannot speak to the chemical analysis of the biochar (XRD etc) I assume this was performed well.

Response: Thanks

Comment 2: The statistics section requires much more detail. Please indicate in the tables which values are different from each other.

Response: More detail information on statistical section were provided in the revised version. 

Comment 3: Recommendations and conclusions are redundant. The paper is not that long and the conclusions are just not needed.

Response: Based on your suggestion, the conclusions section has been deleted and the section of recommendation has been improved in the revised version.

Comment 4: Line 63 – mitigate climate– add change

Response: Thanks, and it has been inserted.

Comment 5: Lines 72-75 These sentences are contradictory as one says there has not been many studies in alkaline soils and then the next sentence sites three. I would just delete the last sentence.

Response: The sentence has been deleted.

Comment 6: Line 77 – delete “on the other hand” (there is no first hand).

Response: it has been deleted. 

Comment 7: Lines 82-87 should be tacked on to the paragraph above it. The last paragraph of the sentence should start “Previous studies…”

Response: Thanks for the reviewer comment and this paragraph has been improved in the revised version.

Comment 8: Line 93 - Replace “the aims of this study were to investigate” to “We investigated”. -this gets rid of redundant words and is in an active voice.

Response: It has been replaced.

Comment 9: Delete line 143-144 – one can just put “Table 1” at the end of the next sentence.

Response: Thanks for the reviewer comment and the mentioned sentence has been deleted.

Comment 10: All tables need to be formatted similarly and to the journal’s standards.

Response: All tables has been formatted according to the journal’s standards.

Comment 11: Line 152- what is basicity (do you mean pH?)

Response: Yes, the basicity reflects on increasing the value of pH. Therefore, word of basicity replaced by pH in the revived version.

Comment 12: Line 209 –This paragraph is very confusing and overly wordy – please simplify

Response: Thanks for the reviewer comment and the mentioned sentence has been rewritten and simplified.

Reviewer #2: 

Comment 1: The production of BC from OMSW is an important area of research

In the section of Material and methods it is not clear if the BC was washed before being mixed with the soil, otherwise some of contaminants were introduced into the soil.

Response: Thanks for the reviewer comment and we agree with you that washed and unwashed biochar may have different behaviour and impacts on soil and plant. In the current study, we used unwashed BC samples. We clarified it in the section of material and methods.

Comment 2: From the beginning you selected alkaline soil and the BC is going to increase that. Did you try some acidic soil which is more suitable for your BC?? Why not from the beginning you selected to compare acidic vs alkaline soil?? Your BC is better suit acidic soil

Response: Because this work was carried out in the arid region where soils have high pH value and high CaCO3 content and there are no acidic soils under our condition. Therefore, our team of the biochar group is interested in studying biochar as a soil amendment under alkaline soil conditions.

Comment 3: The Zeta potential was not measured and data should be supplied, otherwise it is difficult to explain why for example the nutrients such as Ca, Mg and P and others were limited to the plants. 

Response: Based on your valuable comment, we measured the values of zetal potential for BC samples and inserted in the revised version. However, we would like to clarify that there are other reasons could be responsible for nutrients limiting to plants (such as alkalinity) due to biochar application, especially with increasing pyrolysis temperature, as we found that the values of zeta potential tented to decrease with increasing pyrolysis temperature. 

Comment 4: The Tables 1 and 2 no mention how many replicates, SD and statistical analysis such as LSD are available. You are talking about significance, but I can't see that.

Response: Thanks for the reviewer comment and the ±SD values are incorporated in Tables 1 and 2 because the part of biochar characterization carried out in duplicate. We clarified it in the section of material and methods. 

Comment 5: I don't see data concerning N content in all soils especially you used NPK in the INF treatment.

Response: Unfortunately, the soil N availability and its content in plant shoots were not measured in the present work. However, we would like to clarify that the aim of this study was to focus on the effects of biochar on essential nutrients that mainly face problems (especially that subjected to fixation such as P and micronutrients) in alkaline soils and can change with changing soil pH. In addition, in the current study, the recommended dose of NPK was applied in this study because this is the common practice used by the farmers. 

Comment 6: You see from the results (for example Table 5) that when you have increased the soil amendment from 1% to 3% you see increase in the negative effects on plant parameters which means you have contamination in your BC which means you should wash it!!! 

Might be that your supplied BC have affected your soil properties and plant growth as well. Accordingly, your results are indicating that BC supplementation didn't improve plant growth parameters.

Response: Thanks for the reviewer comment and we agree with you that washed and unwashed biochar may have different behaviour and impacts on soil and plant. In the current study, we used unwashed BC samples. In term of applying BC to agricultural soils, unwashed biochar would be mostly added because most of the fresh and unwashed biochars were a source of nutrients. And, our findings suggest that application of OMSW-derived BCs may not be able to provide sufficient nutrients to enhance plant growth. However, based in your comment, this point regarding washed BC will be taken into our account in the future work. Therefore, we referred to this point regarding washed biochar in section of recommendations and suggestions for future trials in the revised version of our manuscript. Thanks again.

---

## [Decision Letter · Decision Letter 1]

15 Jun 2020

Potential short-term negative versus positive effects of olive mill-derived biochar on nutrient availability in a calcareous loamy sand soil

PONE-D-20-11578R1

Dear Dr. Al-wabel,

We’re pleased to inform you that your manuscript has been judged scientifically suitable for publication and will be formally accepted for publication once it meets all outstanding technical requirements.

Kind regards,

Andrés Rodríguez-Seijo, PhD

Academic Editor

PLOS ONE

Additional Editor Comments (optional):

Reviewers' comments:

Reviewer's Responses to Questions

**Comments to the Author**

1. If the authors have adequately addressed your comments raised in a previous round of review and you feel that this manuscript is now acceptable for publication, you may indicate that here to bypass the “Comments to the Author” section, enter your conflict of interest statement in the “Confidential to Editor” section, and submit your "Accept" recommendation.

Reviewer #1: All comments have been addressed

Reviewer #2: All comments have been addressed

2. Is the manuscript technically sound, and do the data support the conclusions?

Reviewer #1: Yes

Reviewer #2: Yes

3. Has the statistical analysis been performed appropriately and rigorously? 

Reviewer #1: Yes

Reviewer #2: Yes

4. Have the authors made all data underlying the findings in their manuscript fully available?

Reviewer #1: Yes

Reviewer #2: Yes

5. Is the manuscript presented in an intelligible fashion and written in standard English?

Reviewer #1: Yes

Reviewer #2: Yes

6. Review Comments to the Author

Reviewer #1: They did a great job revising the manuscript. I am require to write 100 characters so I am just typing something.

Reviewer #2: AS I mentioned in my comments, it is important to wash the BC before being used to cultivate plants. I understood that you didn't wash the BC before planting in the pots which may cause negative effects on plants. The main points/comments I have raised were corrected which make the paper more suitable for publications.

7. PLOS authors have the option to publish the peer review history of their article (what does this mean?). If published, this will include your full peer review and any attached files.

Reviewer #1: No

Reviewer #2: Yes: Prof. Hassan Azaizeh, Tel Hai Academic College, Upper Galilee 12210, Israel

---

## [Editor Report · Acceptance letter]

18 Jun 2020

PONE-D-20-11578R1 

Potential short-term negative versus positive effects of olive mill-derived biochar on nutrient availability in a calcareous loamy sand soil 

Dear Dr. Al-Wabel:

I'm pleased to inform you that your manuscript has been deemed suitable for publication in PLOS ONE. Congratulations! Your manuscript is now with our production department. 

Kind regards, 

on behalf of

Dr. Andrés Rodríguez-Seijo 

Academic Editor

PLOS ONE